# Molecular Modelling of Optical Biosensor Phosphorene-Thioguanine for Optimal Drug Delivery in Leukemia Treatment

**DOI:** 10.3390/cancers14030545

**Published:** 2022-01-21

**Authors:** Shabeer Ahmad Mian, Shafqat Ullah Khan, Akbar Hussain, Abdur Rauf, Ejaz Ahmed, Joonkyung Jang

**Affiliations:** 1Department of Physics, University of Peshawar, Peshawar 25120, Pakistan; shafqat@uop.edu.pk (S.U.K.); akbar_hussain321@uop.edu.pk (A.H.); raufabdur56@yahoo.com (A.R.); 2Department of Physics, Abdul Wali Khan University Mardan, Mardan 23200, Pakistan; ejazahmed@awkum.edu.pk; 3Department of Nano Energy Engineering, Pusan National University, Pusan 46241, Korea

**Keywords:** anticancer drug, physisorption, biodegradability, electrophilicity index, chemical potential, optical biosensor

## Abstract

**Simple Summary:**

Nanocarriers have been used to solve the problems associated with conventional antitumor drug delivery systems, including no specificity, severe side effects, burst release and damaging the normal cells. It improves the bioavailability and therapeutic efficiency of antitumor drugs, while providing preferential accumulation at the target site. Various 2D nanomaterials such as graphene, MoS2, and WSe2 have been used as nanocarrier. The recent discovery of phosphorene has introduced new possibilities in designing a sensible drug delivery system, due to low cytotoxicity, biocompatibility, high surface to volume ratio, which can increase its drug loading capacity. The biodegradation of phosphorene inside the human body produces non-toxic intermediates, like phosphate. Phosphate is necessary for the formation of bone and teeth. Phosphate is also used by the cell for energy, cell membranes, and DNA (deoxyribonucleic acid). Therefore, phosphorene nanocarrier is not harmful, especially for the treatment of cancer in vivo applications.

**Abstract:**

Thioguanine is an anti-cancer drug used for the treatment of leukemia. However, thioguanine has weak aqueous solubility and low biocompatibility, which limits its performance in the treatment of cancer. In the present work, these inadequacies were targeted using density functional theory-based simulations. Three stable configurations were obtained for the adsorption of thioguanine molecules on the phosphorene surface, with adsorption energies in the range of −76.99 to −38.69 kJ/mol, indicating physisorption of the drug on the phosphorene surface. The calculated bandgap energies of the individual and combined geometries of phosphorene and thioguanine were 0.97 eV, 2.81 eV and 0.91 eV, respectively. Owing to the physisorption of the drug molecule on the phosphorene surface, the bandgap energy of the material had a direct impact on optical conductivity, which was significantly altered. All parameters that determine the potential ability for drug delivery were calculated, such as the dipole moment, chemical hardness, chemical softness, chemical potential, and electrophilicity index. The higher dipole moment (1.74 D) of the phosphorene–thioguanine complex reflects its higher biodegradability, with no adverse physiological effects.

## 1. Introduction

The development of effective drug delivery systems has attracted tremendous attention in the last few decades because of their high drug delivery potential to targeted tissues without causing any detrimental effects [1]. Therefore, drug delivery systems have been designed to release drugs in a controlled manner at the desired site to improve their bioavailability and pharmacological properties [2]. To further improve their performance, nanoengineering has been introduced in the pharmaceutical sphere, and its applications are growing rapidly [3]. Nanocarriers have garnered significant interest in the field of nanomedicine [4]. These nanostructures are developed such that they can interact individually with infected cells and aid their direct treatment, which reduces the side effects of the drug on healthy cells, tissues, and organs [5].

Various two-dimensional (2D) nanomaterials such as molybdenum disulfide (MoS_2_) [6], tungsten diselenide (WSe_2_) [7], hexagonal boron nitride (h-BN) [8], and graphene [9] are used as drug delivery carriers because of their unique physical and chemical properties. However, among different 2D materials, phosphorene is a promising alternative drug delivery carrier material to graphene [10]. Owing to its relatively low cytotoxicity and high biocompatibility, phosphorene has introduced possibilities for designing novel candidates for drug delivery [11]. The biocompatibility of 2D materials depends on their size, material, and concentration; in addition, 2D nanomaterials possess excellent biocompatibility [12]. Phosphorene is substantially less cytotoxic than graphene but more cytotoxic than MoS_2_ and WSe_2_ [13]. Therefore, phosphorene is an excellent candidate for photodynamic therapy, photothermal therapy, bioimaging, and drug delivery systems [14]. It has a higher drug-loading capacity than h-BN and graphene owing to its wrinkled lattice arrangement, which increases its surface-to-volume ratio [15].

Phosphorene exhibits a layer-dependent bandgap (0.3 eV–1 eV) [16]. This correlates to interesting optical properties, which helps in the colorimetric and fluorescent analysis of different biomolecules. These special characteristics make phosphorene a useful candidate for photothermal treatment, photodynamic treatment, biosensing, bioimaging, and drug delivery system [17]. Owing to its wrinkled lattice arrangement, phosphorene exhibits different anisotropic optical, mechanical, and magnetic properties [18]. Moreover, in comparison with the zigzag direction, the thermal conductivity and Young’s modulus along the armchair direction were less [19].

Phosphorene is also highly biodegradable inside the human body, producing nontoxic intermediates such as phosphates and phosphite. Therefore, it is suitable for biomedical applications for in vivo studies [20]. Other 2D materials may accumulate inside the human body and contribute to their cytotoxicity because graphene requires proper functionalization with oxygen to enable effective clearance. Therefore, phosphorene has many suitable properties for biomedical applications such as photothermal therapy, photodynamic therapy, biosensing, bioimaging, and drug delivery [21].

Many anticancer medicines are effective in the treatment of cancer. However, these anticancer drugs have many inherent limitations in terms of toxicity. For example, thioguanine has numerous side effects, including acute bone marrow suppression, nausea, and hair loss [22]. To enhance the therapeutic effects and minimize the side effects, scientists have primarily focused on nanoscale drug delivery carriers [23,24]. In this study, the ability of phosphorene as a drug delivery carrier for thioguanine to treat cancer was evaluated using the density functional theory (DFT).

## 2. Methods and Materials

A phosphorene monolayer as a nanocarrier was considered for the adsorption of thioguanine drug molecules using DFT simulations [25]. The exchange correlation function was treated using the generalized gradient approximation (GGA) [26], with the Revised Perdew-Burke Ernzerhof (RPBE) [27] functional, for the optimization of all geometries in this study.

A hybrid DFT [28], with 0.01% local density approximation and 0.99% GGA, was used for better characterization of the electronic structures of various materials. The optimized geometries of both the drug molecule and nanocarrier were combined for the adsorption process and simulated using the SIESTA software. An energy cut-off of 200 Ry, and a 3×3×1 k-point sampling was set for geometry optimization, while for electronic properties calculation of the individual and combine optimized geometries a 6×6×6 k-points were used. The pseudo-atomic orbital basis set with double zeta potential was used for all atoms in all configurations [29].

## 3. Results and Discussion

### 3.1. Geometries Analysis

Different orientations of the drug molecule thioguanine were used for its adsorption on the surface of the phosphorene monolayer to obtain the most stable configuration. The relaxed geometries of phosphorene, thioguanine, and the complex are shown in Figure 1. It was observed that all geometries obtained the minimum energy state for thermodynamic stability during the optimization process. The variations in the free energy and volume for phosphorene, thioguanine, and their combined geometries (complex material) with the number of iterations are schematically represented in Figure 2 and Figure 3. It can be observed that the free energy and volume of each system after optimization attain the minimum energy and volume.

### 3.2. Calculation of Adsorption Energy of the Complex

It is important to determine the adsorption energy of a material during its interaction with the surface to evaluate the thermodynamic stability and favorability of its synthesis in the laboratory. The adsorption energy of the thioguanine drug molecule on the phosphorene surface was calculated using the following equation [30,31]:(1)$$E_{\mathrm{adsorption}}={\;E}_{\mathrm{complex}}-{\;E}_{\mathrm{carrier}}-E_{\mathrm{drug}}$$
where E_adsorption_ is the adsorption energy of the complex material, and E_complex_, E_drug_, and E_carrier_ represent the total energy of the complex, thioguanine drug, and phosphorene carrier, respectively. In the investigated configurations, thioguanine was initially placed at different positions on the phosphorene surface, as shown in Figure 4. The calculated adsorption energies of the three investigated configurations are shown in Figure 5. The negative adsorption energies for all cases indicate the thermodynamic stability and favorability of thioguanine adsorption on the phosphorene surface.

Figure 4 depicts all the orientations of the drug molecule physiosorbed on the carrier surface. Among the three different orientations considered, the Configuration3 geometry, shown in Figure 4, is thermodynamically the most stable configuration, attributed to the more negative adsorption energy. Furthermore, it may lead to several intermolecular interactions between the drug molecule and the phosphorene surface.

The calculated adsorption energy for thioguanine on the phosphorene monolayer was consistent with previous reports [32]. Moreover, compared to the adsorption of thioguanine on the surface of graphene oxide, it is thermodynamically more stable. This comparison indicated that phosphorene is a suitable candidate for the delivery of drug molecules.

Therefore, it can be concluded that our investigated adsorption energy value ranges are comparable to the adsorption range of thioguanine on the blue phosphorene surface [33].

### 3.3. Frontier Molecular Orbital Analysis

The highest occupied molecular orbital (HOMO) and lowest unoccupied molecular orbital (LUMO) are also called frontier molecular orbitals (FMOs), which provide interesting information about the reactivity of a molecule [34]. The energy gap between the HOMO and LUMO (E_g_) can be expressed as follows:(2)$$E_{g}=E_{\mathrm{LUMO}}-E_{\mathrm{HOMO}\;}$$

The energy gap E_g_ provides information for understanding the kinetic stability and chemical reactivity of the system. A molecule with a small energy gap is more polarizable and generally exhibits low kinetic stability and high chemical reactivity [35].

The band structures of the phosphorene, thioguanine, and thioguanine–phosphorene complexes are shown in Figure 6. Different computational approaches were used to evaluate the HOMO and LUMO energy levels and energy gaps of the phosphorene, thioguanine, and phosphorene–thioguanine complexes. The results obtained from the GGA with the exchange-correlation authors, RPBE, were consistent with the available results, as shown in Table 1.

The obtained bandgap energy was in agreement with the experimental data. The energy bandgap of the phosphorene–thioguanine complex (0.91 eV) is lower than that of phosphorene (0.97 eV) and thioguanine (2.81 eV), indicating that the complex requires less energy for excitation. The narrow bandgap indicated that the drug was successfully attached to the carrier surface [11]. The FMOs, including the HOMO and LUMO, are indicated as significant parameters for chemical reactions [39]. The energy gap between the HOMO and LUMO is an important factor in determining the electrical transport properties of molecular systems [40].

The interaction between thioguanine and phosphorene can be further investigated by calculating projected density of states (PDOS) [41]. The PDOS for phosphorene, thioguanine, and phosphorene/thioguanine are shown in Figure 7a–c. The PDOS plots reveal that the energies of HOMO orbital in thioguanine increased from −4.34 eV to −3.91 eV after their attachment to the phosphorene surface, whereas the energy of LUMO orbital decreased from −1.44 eV to −3.00 eV, as shown in Table 2. The PDOS in Figure 7 demonstrates that some new intermediate states appear owing to the adsorption of thioguanine on the phosphorene surface and the intermediate states narrow the bandgap energy. This ultimately improves the conductivity of the material.

### 3.4. Optical Conductivity

The calculated electronic structures revealed that the lower bandgap energy was due to the creation of intermediate permissible energy states of the adsorbed species on the surface. The narrow bandgap energy of the complex material significantly affects its optical conductivity according to the following relation:(3)$$\sigma =\frac{\alpha nc}{4\pi }\;$$
where “*n*” is the refractive index “α” is the absorption coefficient and “*c*” is the speed of light. The above Equation (3) demonstrates that optical conductivity has a direct dependence on optical absorption. The optical conductivities of the phosphorene, thioguanine, and thioguanine–phosphorene complexes are shown in Figure 8a. The additional molecular energy levels after molecule adsorption on the phosphorene surface increased the HOMO (−3.91 eV) energy while decreasing the LUMO (−3.00 eV) energy, which was the main reason for narrowing the energy bandgap of the complex material responsible for the charge-transfer process. A material with a lower energy gap is more conductive and reactive than a material with a higher energy gap. It can be seen from Table 1 that the bandgap energy of thioguanine after the adsorption process on the phosphorene surface was significantly reduced.

Thus, a notable change occurred in the conductivity and reactivity of the system after the linkage of drugs on the surface of phosphorene. These results also indicated that the thioguanine drug was successfully attached and adsorbed on the surface of phosphorene. The absorption spectra of thioguanine, phosphorene, and their complexes are shown in Figure 8b.

The phosphorene–thioguanine complex absorbs at a much lower energy (0.91 eV) than thioguanine (2.81 eV), which allows a broad absorption spectrum (248–1362 nm). The absorption edge at energy 0.91 eV in the absorption spectra of the complex material elucidates the calculated HOMO and LOMO positions. This correlation suggests that the excitation at 0.91 eV corresponds to the excitation from the HOMO to LUMO, which in turn is the transfer of charge from the drug to phosphorene.

### 3.5. Polarity of the Molecules

The dipole moment of a molecule is a major factor directly affecting its solubility [42]. Compounds with higher dipole moments exhibit increased solubility in polar solvents such as water. It can be seen from Table 2 that the dipole moment of the complex was higher than that of the drug molecule. Therefore, the solubility of thioguanine drug significantly increased after binding with phosphorene, indicating higher biodegradability of the complex material without any harmful physiological effects.

### 3.6. Quantum Molecular Descriptors

The quantum chemical parameters of phosphorene, thioguanine, and their complexes provide information about the chemical reactivity of the molecules, as summarized in Table 2. The chemical behavior of the studied molecules can be predicted using these parameters [32].
(4)$$\mathrm{Ionization}\;\mathrm{potential}\;\mathrm{IP}=-E_{\mathrm{HOMO}}\;$$
(5)$$\mathrm{Electron}\;\mathrm{affinity}\;\mathrm{EA}=-E_{\mathrm{LUMO}\;}$$

The electronegativity (χ) is the degree to which an element tends to gain electrons and form negative ions in a chemical reaction [43].
(6)$$\chi =\frac{\left(\mathrm{IP}+\mathrm{EA}\right)}{2}$$

The escaping tendency of an electron from an equilibrium state is represented by the chemical potential (µ).
(7)µ=−χ

The resistance of a chemical species to changes in its electronic structure is the chemical hardness (η). A positive or negative change in the chemical hardness affects stability and reactivity.
(8)$$\eta =\frac{\mathrm{IP}-\mathrm{EA}}{2}$$

The reciprocal of chemical hardness is the softness (S), and it is used to measure the degree of chemical reactivity, which represents the tendency of an atom or a group of atoms to receive electrons.
(9)$$S=\frac{1}{2\eta }$$

The electrophilicity index (ω) can be used to estimate the reactivity of a molecule [44]:(10)$$\omega =\frac{{\text{µ}}^{2}}{2\eta }$$

The amount of charge transfer between thioguanine (drug) and phosphorene (carrier) can be calculated as:(11)$$\Delta N=\frac{1}{2}\frac{\left({\text{µ}}_{d}-{\text{µ}}_{c}\right)}{\left(\eta_{d}-\eta_{c}\right)}$$

A positive value of ΔN indicates charge transfer from the drug to phosphorene, whereas a negative value indicates charge transfer from the carrier phosphorene to the drug. The ionization potential of the phosphorene–thioguanine complex is lower than that of the thioguanine drug molecule, indicating that the complex requires more energy than the thioguanine molecule to become a cation. From Table 2, it can be seen that the phosphorene–thioguanine complex is highly reactive owing to its high electron affinity. This high electron affinity value indicates an electrophilic structure that forms a stable negative ion.

An important application of electronegativity is the prediction of the polar nature of the resulting structure after adsorption. This confirmed the low hardness, higher softness, and higher electrophilicity index. Hardness is an important characteristic for measuring molecular stability and reactivity, and a decrease in hardness is a sign of a narrow bandgap that lowers the resistance of species to lose an electron. The adsorption of thioguanine results in the softness of the phosphorene. A high value of the electrophilicity index (ω) indicates a higher electrophilic character of the molecules. Therefore, the value of the electrophilicity index increased owing to an increase in charge transfer from the drug to phosphorene. As the value of (ΔN) is positive, phosphorene acts as an electron acceptor. The results obtained from ΔN and ω were in agreement with each other [45].

## 4. Conclusions

The interaction between phosphorene and thioguanine, including their adsorption energy and electronic properties, was examined using density functional theory simulations. These results indicate that thioguanine molecules are physisorbed. In the most stable configuration, the thioguanine molecule prefers a parallel orientation to achieve maximum interaction with surface atoms. The optical conductivity and reactivity of the drug molecule were significantly altered after binding to the phosphorene surface, which was attributed to its lower bandgap. Charge transfer occurs from the drug molecule to the phosphorene nanocarrier due to the decrease in bandgap, ionization potential, and chemical hardness while increasing the electron affinity, electrophilicity index, and negative molecular charge transfer of the complex. The higher dipole moment (1.74 D) of the phosphorene–thioguanine complex suggests higher biodegradability, with no harmful physiological effects. In conclusion, these results demonstrate that phosphorene can be used as an appropriate nanocarrier for the delivery of thioguanine drug molecules.

## Figures and Tables

**Figure 1 cancers-14-00545-f001:**
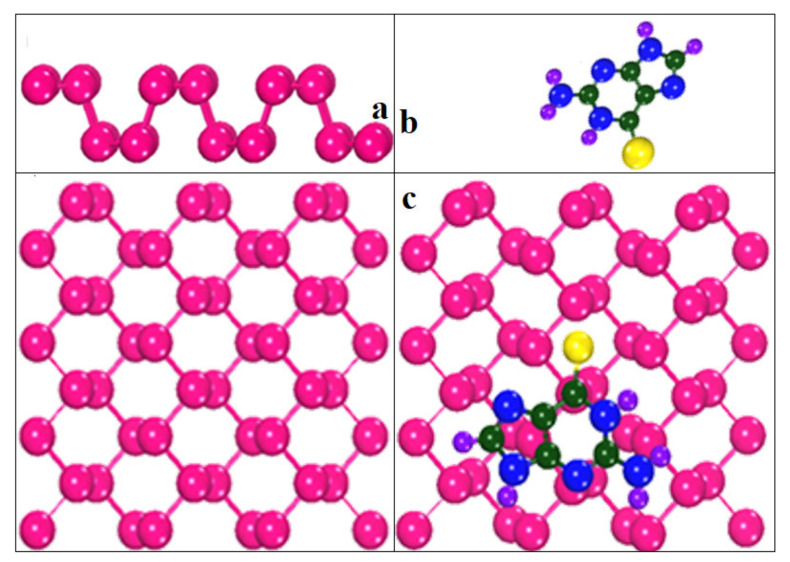
Optimized geometries of (**a**) phosphorene (**b**) thioguanine (**c**) phosphorene–thioguanine complex.

**Figure 2 cancers-14-00545-f002:**
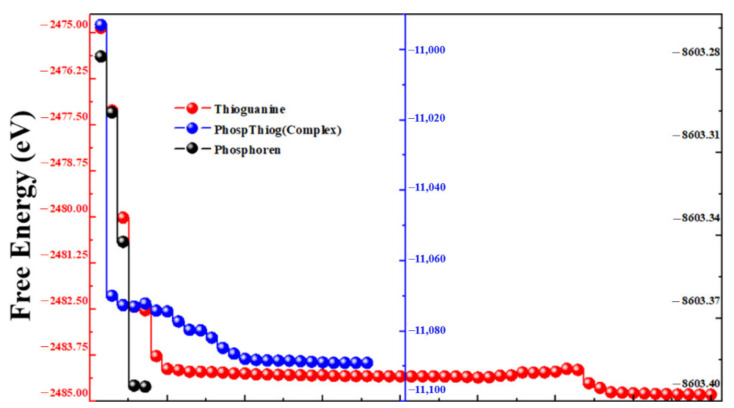
Free energy versus the number of iterations of phosphorene, thioguanine and phosphorene–thioguanine complex.

**Figure 3 cancers-14-00545-f003:**
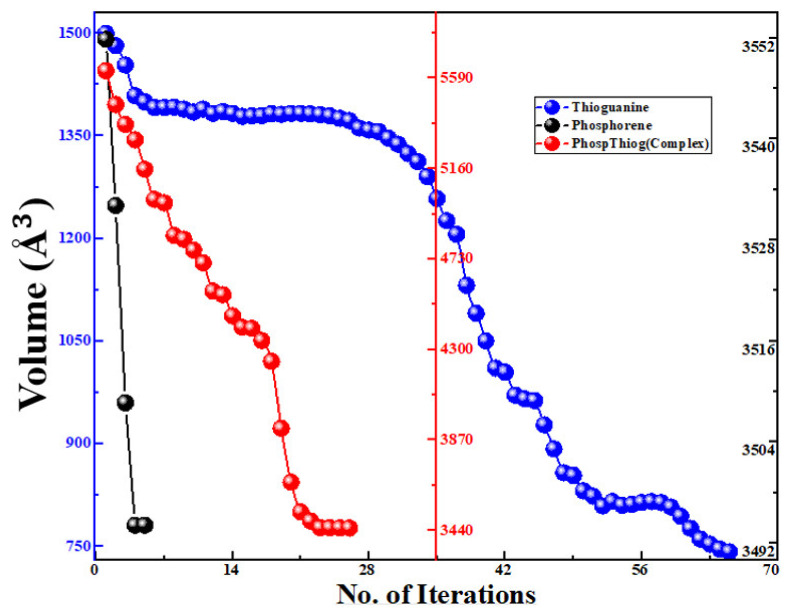
Volume versus the number of iterations of phosphorene, thioguanine, and phosphorene–thioguanine complex.

**Figure 4 cancers-14-00545-f004:**
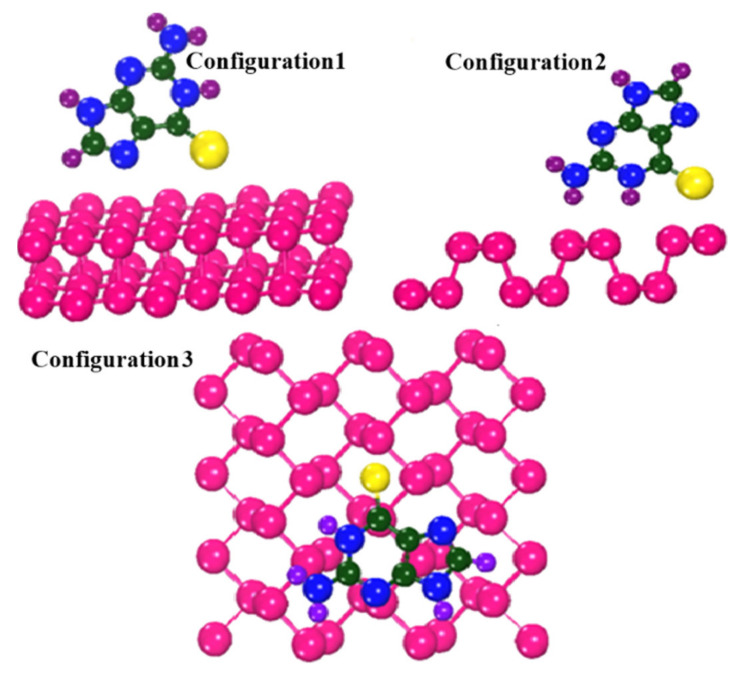
Different configurations of phosphorene–thioguanine complex.

**Figure 5 cancers-14-00545-f005:**
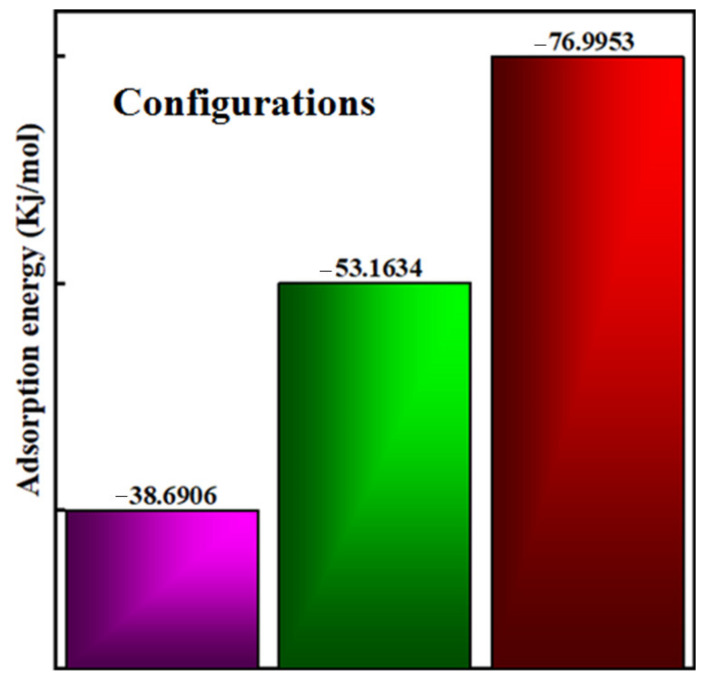
Calculated adsorption energy of the complex of different configurations.

**Figure 6 cancers-14-00545-f006:**
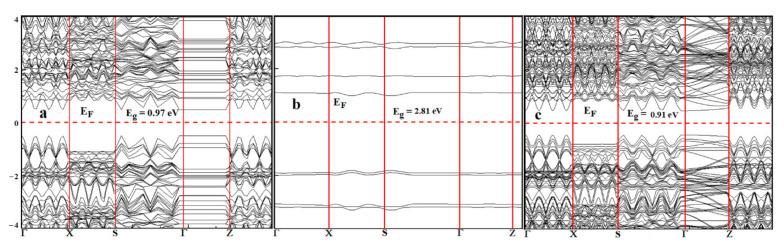
Band structure of (**a**) phosphorene, (**b**) thioguanine, and (**c**) phosphorene–thioguanine complex.

**Figure 7 cancers-14-00545-f007:**
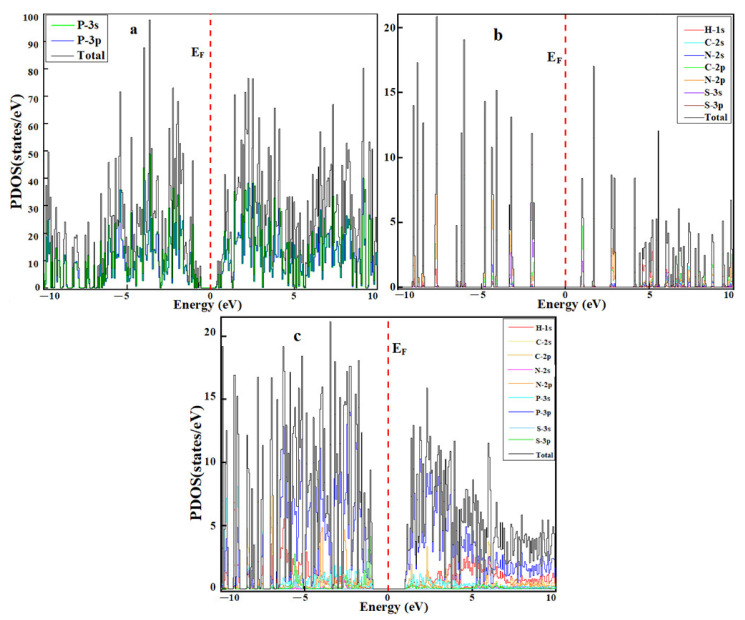
Projected density of states of (**a**) phosphorene, (**b**) thioguanine, (**c**) phosphorene–thioguanine complex.

**Figure 8 cancers-14-00545-f008:**
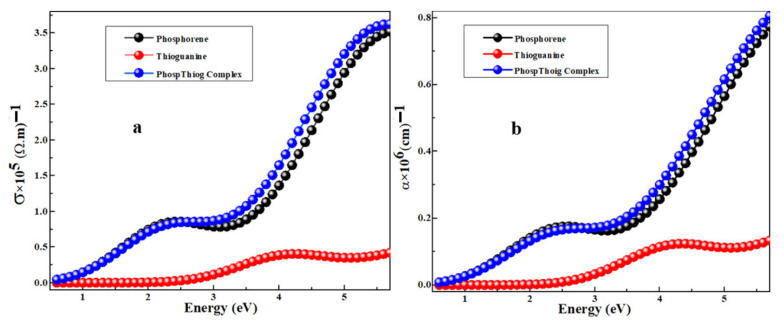
Analysis of (**a**) optical conductivity and (**b**) optical absorption spectra of phosphorene, thioguanine, and their complex structure.

**Table 1 cancers-14-00545-t001:** The calculated values of HOMO, LUMO energy level, and energy gap (E_g_) all in (eV) of phosphorene, thioguanine, and their complex.

Material	Method	E_LUMO_ (eV)	E_HOMO_ (eV)	E_g_ (eV)	Experiment	Other DFT
Phosphorene	RPBE	−2.90	−3.88	0.97	1–1.45 [16,36]	0.76–0.92 [37]
	PBE	−3.01	−3.95	0.94		
	revPBE	−2.91	−3.88	0.97		
	Hybrid	−3.08	−4.05	0.97		
Thioguanine	RPBE	−1.23	−4.06	2.83	3.58 [38]	7.20–0.14 [38]
	PBE	−1.36	−4.18	2.82		
	revPBE	−1.24	−4.06	2.82		
	Hybrid	−1.44	−4.34	2.81		
Phosp/Thiog	RPBE	−2.80	−3.71	0.91	-	-
	PBE	−2.91	−3.79	0.88		
	revPBE	−2.81	−3.72	0.91		
	Hybrid	−3.00	−3.91	0.91		

**Table 2 cancers-14-00545-t002:** Calculated values of dipole moment (D), electron affinity, ionization potential, electronegativity, chemical potential, chemical hardness, chemical softness, electrophilicity index (all in eV), and molecular charge transfer of phosphorene, thioguanine, and their complex.

Chemical Properties	Phosphorene	Thioguanine	PT Complex
Dipole moment (D)	0.0005	0.28	1.74
Electron affinity	3.08	1.44	3.00
Ionization potential (eV)	4.05	4.24	3.91
Electronegativity (eV)	3.56	2.84	3.45
Chemical potential (eV)	−3.56	−2.84	−3.45
Chemical hardness	0.48	1.40	0.45
Chemical softness (eV)^−1^	1.04	0.35	1.11
Electrophilicity index (eV)	13.20	2.88	13.22
Molecular charge transfer (ΔN)			0.39

## Data Availability

Even though all the data is included in this publishable work but if someone is interested, we are ready to guide them.

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
