# Peer review of "Molecular Modelling of Optical Biosensor Phosphorene-Thioguanine for Optimal Drug Delivery in Leukemia Treatment"

_cancers, 2022, doi:10.3390/cancers14030545_

Round 1

Reviewer 1 Report

The paper studies the interaction of thioguanine with phosphorene as a possible drug-delivery mechanism. There are some issues that need to be addressed before publication.

Figures 2 and 3 are not necessary; the values at convergence can be shown in a table. However, the way the volume is defined need to be specified. Also, is the quantity called Free Energy the same as Ecomplex in Eq. (1)? If yes the notation has to be made consistent; if not it has to be explained what the difference is between the two.

What geometry was used for Edrug in Eq .(1)?

Line 122: ‘complex3’ should be changed to ‘configuration 3’ to stay consistent with the figure.

In Lines 126-127 it is said that “ The calculated adsorption energy for thioguanine drug on phosphorene monolayer is in good agreement with the available results[32]” but Ref 32 is on a similar but different system. So instead of ‘good agreement with’ I suggest saying ‘is consistent with’.

Finally, a discussion of the drug delivery should address the dissociation of the drug from the carrier. In the present case, the three conformations discussed had different adsorption energies. While the weakest one being commensurable to the binding free energy of a good drug, the most strongly bound is much bound stronger than most drugs.

In addition, it is not clear (at least to this reviewer) if the issue whether the interaction of thioguanine with phosphorene can result in a chemical reaction between the two was examined.

Author Response

  • Figures 2 and 3 are not necessary; the values at convergence can be shown in a table. However, the way the volume is defined need to be specified. Also, is the quantity called Free Energy the same as Ecomplex in Eq. (1)? If yes the notation has to be made consistent; if not it has to be explained what the difference is between the two.
  • Reply:  

    Figure 2 and Figure 3, represents the thermodynamic stability of a material. As we can see from both figures that before optimization; Phosphorene, thioguanine, and phosphorene-thioguanine have higher volumes while their free energies have low values. But after optimization, their volume becomes decrease, while free energy increase in the negative direction. This indicates that all the materials gain stability during optimization. Because mostly thermodynamic reactions occur at the ground state.

  • Is the quantity free energy of the complex is the same as Ecomplex?

  • Reply: Yes free energy of the complex is the same as Ecomplex.

  • What geometry was used for Edrug?

  • Reply: For Edrug, we used the geometry of thioguanine shown in figure1 (b)

  • Line 122: ‘complex3’ should be changed to ‘configuration 3’ to stay consistent with the figure.
  • Reply: corrected
  • In Lines 126-127 it is said that “ The calculated adsorption energy for thioguanine drug on phosphorene monolayer is in good agreement with the available results[32]” but Ref 32 is on a similar but different system. So instead of ‘good agreement with’ I suggest saying ‘is consistent with’.
  • Reply: corrected to consistent
  • Finally, a discussion of the drug delivery should address the dissociation of the drug from the carrier. In the present case, the three conformations discussed had different adsorption energies. While the weakest one being commensurable to the binding free energy of a good drug, the most strongly bound is much bound stronger than most drugs.
  • Reply: the three conformations discussed had different adsorption energies in the phsisoprtion state not in the chemisorption. confirmation three is more suitable as it is lying in the XY-plane of the surface and thus having high free energy.
  • In addition, it is not clear (at least to this reviewer) if the issue whether the interaction of thioguanine with phosphorene can result in a chemical reaction between the two was examined.
  • Reply: No chemical Reaction between the two

Reviewer 2 Report

The manuscript “Molecular Modelling of Optical Biosensor Phosphorene-Thioguanine for the Detection of Leukemia Biomarker” is submitted to the Cancer Journal.

Generally, I found the manuscript very interesting. However, several suggestions and clarifications are suggested:

  • Please be constant with the references: in the text, some references are after the dot and some before.
  • The same applied to the complex writing “Phosphorene-Thioguanine” or “Phosphorene/Thioguanine” or “phosphorene-thioguanine”. Chose only one way of writing.
  • Please check the punctuations. Often “.” is together with “,”: pages 171, 172, 227, 243.
  • Page 60, “intermediates such as phosphates and phosphate”. What is the difference?
  • Also, I would recommend changing the Title of the manuscript because in the manuscript, it is written only about drug delivery systems. There is no information about the sensor.

Author Response

  • Please be constant with the references: in the text, some references are after the dot and some before.
  • Reply: corrected
  • The same applied to the complex writing “Phosphorene-Thioguanine” or “Phosphorene/Thioguanine” or “phosphorene-thioguanine”. Chose only one way of writing.
  • Reply: Corrected
  • Please check the punctuations. Often “.” is together with “,”: pages 171, 172, 227, 243.
  • Reply: Corrected
  • Page 60, “intermediates such as phosphates and phosphate”. What is the difference?
  • Reply: Corrected "phosphates and phosphite”
  • Also, I would recommend changing the Title of the manuscript because in the manuscript, it is written only about drug delivery systems. There is no information about the sensor.
  • Reply: Title Modified